# Inhibitory Effects of Vandetanib on Catecholamine Synthesis in Rat Pheochromocytoma PC12 Cells

**DOI:** 10.3390/ijms26146927

**Published:** 2025-07-18

**Authors:** Yoshihiko Itoh, Kenichi Inagaki, Tomohiro Terasaka, Eisaku Morimoto, Takahiro Ishii, Kimitomo Yamaoka, Satoshi Fujisawa, Jun Wada

**Affiliations:** Department of Nephrology, Rheumatology, Endocrinology and Metabolism, Graduate School of Medicine, Dentistry and Pharmaceutical Sciences, Okayama University, 2-5-1 Shikata-cho, Okayama 700-8558, Japan; itoh.lut65b@gmail.com (Y.I.); terasakat@okayama-u.ac.jp (T.T.); fujisawa_sa@okayama-u.ac.jp (S.F.); junwada@okayama-u.ac.jp (J.W.)

**Keywords:** tyrosine kinase inhibitor, multiple endocrine neoplasia type 2, paraganglioma, RET, ERK, AKT

## Abstract

Gain-of-function gene alterations in rearranged during transfection (RET), a receptor tyrosine kinase, are observed in both sporadic and hereditary medullary thyroid cancers (MTCs) and pheochromocytomas and paragangliomas (PPGLs). Several tyrosine kinase inhibitors (TKIs) that target RET have been proven to be effective on MTCs and PCCs. Recently, TKIs, namely, sunitinib and selpercatinib, which were clinically used to target PPGLs, have been reported to decrease catecholamine levels without reducing tumor size. Our clinical case of metastatic medullary thyroid cancer, which is associated with RET mutations undergoing treatment with vandetanib, also suggests that vandetanib can decrease catecholamine levels. Therefore, we investigated the effect of vandetanib, a representative multi-targeted TKI for RET-related MTC, on cell proliferation and catecholamine synthesis in rat pheochromocytoma PC12 cells. Vandetanib reduced viable cells in a concentration-dependent manner. The dopamine and noradrenaline levels of the cell lysate were reduced in a concentration-dependent manner. They also decreased more prominently at lower concentrations of vandetanib compared to the inhibition of cell proliferation. The RNA knockdown study of Ret revealed that this inhibitory effect on catecholamine synthesis is mainly mediated by the suppression of RET signaling. Next, we focused on two signaling pathways downstream of RET, namely, ERK and AKT signaling. Treatment with vandetanib reduced both ERK and AKT phosphorylation in PC12 cells. Moreover, both an MEK inhibitor U0126 and a PI3K/AKT inhibitor LY294002 suppressed catecholamine synthesis without decreasing viable cells. This study in rat pheochromocytoma PC12 cells reveals the direct inhibitory effects of vandetanib on catecholamine synthesis via the suppression of RET-ERK and RET-AKT signaling.

## 1. Introduction

Rearranged during transfection (RET), a receptor tyrosine kinase, is crucial for normal development, maturation, and maintenance of several organs, including neural and genitourinary tissues [1]. The expression levels of RET are high in early embryogenesis and comparatively low in normal adult tissues [2]. In endocrine tissues, RET is expressed in thyroid C cells and adrenal chromaffin cells [3]. Abnormally activated RET signaling is recognized as a critical inducer of several human cancers. Germline-activating *RET* variants cause multiple endocrine neoplasia type 2 (MEN2), which is a hereditary cancer syndrome associated with medullary thyroid cancer (MTC) in nearly 100% of patients and pheochromocytoma (PCC) in approximately 50% of patients [4]. Somatic *RET* pathogenic variants are also detected in approximately 60% of sporadic MTC cases [5]. Furthermore, various somatic genetic rearrangements that lead to chimeric RET oncoproteins are found in sporadic papillary thyroid cancer, non-small-cell lung cancer (NSCLC), and, rarely, other cancer types [6,7].

Because of the roles of RET in the development and progression of these cancers, RET has been an important target for therapeutic intervention. RET is structurally similar to other receptor tyrosine kinases and has high homology in its functional domains, including the ATP-binding sequences, and in key conserved motifs [1,8]. Thus, RET can be inhibited by several small-molecule tyrosine kinase inhibitors (TKIs) including vandetanib, cabozantinib, sunitinib, sorafenib, and lenvatinib, which have been developed against other tyrosine kinase receptors such as vascular endothelial growth factor receptor (VEGFR), platelet-derived growth factor receptor (PDGFR), and epidermal growth factor receptor (EGFR). However, each TKI has different inhibitory activities against its various targets and consequently exerts distinct therapeutic effects on RET-related cancers [9].

PCC and paraganglioma, collectively referred to as PPGL, are neuroendocrine tumors that arise from neural crest-derived cells of the adrenal medulla and extra-adrenal paraganglia, respectively. PPGLs are considered to have the highest degree of heritability of any human tumor type. More than one-third of PPGLs are associated with inherited cancer susceptibility syndromes. According to The Cancer Genome Atlas (TCGA), PPGLs are categorized into three main molecular subgroups linked to discrete driver genes: cluster 1 (*SDHA*, *SDHB*, *SDHC*, *SDHD*, *SDHAF2*, *FH*, *VHL*, *EPAS1*, and *EGLN1*), cluster 2 (*RET, NF1, TMEM127, MAX, HRAS, FGFR1,* and *MET*), and cluster 3 (*CSDE1* or *MAML3*) (13). Tumors categorized in cluster 2 exhibit abnormal activation of the phosphatidylinositol 3-kinase (PI3K)/AKT (protein kinase B) and Ras/Raf/MEK/extracellular signal-regulated kinase (ERK) signaling pathways [10]. RET-related PCCs belong to this cluster. The risk of metastasis of cluster 2 tumors is relatively low (≤4%) compared with cluster 1 tumors [11,12]. Retrospective observation and clinical trials have indicated that TKIs including sunitinib and cabozantinib could stabilize and decrease the size of some metastatic PPGLs [13,14,15]. Moreover, it was reported that certain TKIs, sunitinib and selpercatinib, which were clinically used to target PPGLs, decreased catecholamine levels in some patients without reducing tumor size [16,17]. Though these TKIs can inhibit not only VEGFR and EGFR but also RET, the effectiveness of RET signaling inhibition in PPGLs remains unclear. Vandetanib has recently been approved for the treatment of locally advanced and metastatic MTC, which is associated with MEN2. Our clinical case of metastatic MTC undergoing treatment with vandetanib also suggests that vandetanib can continually decrease catecholamine levels [18]. During treatment of unresectable PPGLs, cardiovascular adverse effects associated with increased catecholamine release must be avoided. Therefore, we investigated the effect of vandetanib, a representative TKI for RET-related MTC, on catecholamine synthesis and cell proliferation through RET-specific manners using rat pheochromocytoma PC12 cells, which do not express a pathogenic variant of RET but do express that of another cluster 2 gene MAX.

## 2. Results

### 2.1. Effects of Vandetanib on Cell Proliferation and Catecholamine Synthesis in PC12

Vandetanib is known to inhibit RET, VEGFR2, VEGFR3, and EGFR, encoded by Ret, Kdr, Flt4, and Egfr, respectively. To demonstrate the presence of these target tyrosine kinase receptors in PC12 cells, mRNA expressions were examined by reverse transcriptase PCR analysis (Figure 1A and Appendix A). Ret and Egfr, which were reported to be expressed in PPGL in a previous report [19], were clearly detected, whereas Flt4 was considerably less expressed, and Kdr was not detected in PC12 cells. The mRNA expression levels of these receptors were evaluated by quantitative PCR; mRNA expression of Ret and Egfr was predominantly observed in PC12 (Figure 1B), while that of Flt4 was low. 

To investigate the influence of vandetanib on tumor growth in PPGL, we evaluated the viability of PC12 cells treated with various concentrations of vandetanib by measuring cell viability using CellTiter 96^®^ (Figure 2). The number of viable PC12 cells was reduced by treatment with vandetanib in a concentration-dependent manner, with significant reductions observed at higher concentrations above 1000 nmol/L vandetanib. The concentration level of 1000 nmol/L vandetanib is consistent with the dose required to completely inhibit DNA synthesis in patient-derived MTC cells [20].

To examine the effect of vandetanib on catecholamine synthesis in PPGL cells, we assessed the concentration of dopamine (Figure 3A) and noradrenaline (Figure 3B) of PC12 cell lysates treated with vandetanib. Both dopamine and noradrenaline levels in cell lysates were decreased by treatment with vandetanib in a concentration-dependent manner. Treatment with 100 nmol/L of vandetanib decreased dopamine and noradrenaline synthesis by 22.7% and 14.1%, respectively. Notably, inhibition of catecholamine synthesis by vandetanib occurred at lower concentrations than that of cell viability in PC12 cells. The inhibitory effects of vandetanib on dopamine and noradrenaline synthesis did not change after correction for total protein content in cell lysates (Figure 3C,D).

### 2.2. Effects of RET Knockdown on Catecholamine Synthesis in PC12 Cells

To identify the key targets of vandetanib during execution of its inhibitory effects on catecholamine synthesis, knockdown of the candidate tyrosine kinase receptor genes that were expressed in PC12 cells (*Ret*, *Flt4*, and *Egfr*) was performed with siRNA transfection. The knockdown efficiency of each siRNA transfection was confirmed by quantitative RT-PCR (Figure 4A). The expression levels of tyrosine kinase receptor genes were decreased as expected after transfection with siRET, siFLT4, and siEGFR by 44.4%, 29.6%, and 51.5%, respectively.

Next, we examined the effects of knockdown of these tyrosine kinase receptors on catecholamine synthesis. As shown in Figure 4B,C, *Ret* silencing reduced intracellular dopamine and noradrenaline levels by 28.8% and 31.4%, respectively. In contrast to what was observed with knockdown of *Ret*, knockdown of *Flt4* or *Egfr* did not significantly affect catecholamine synthesis. *Egfr* was expressed abundantly in PC12 cells, and the knockdown efficiency of *Egfr* was satisfactory, but surprisingly, catecholamine levels remained unchanged. RET and EGFR have a similar cell signaling pathway, but only EGFR involves the Janus kinase/signal transducers and activators of transcription (JAK/STAT) pathway. This could explain why only RET knockdown decreased catecholamine synthesis, while catecholamine production thorough the JAK/STAT pathway was maintained under EGFR signaling. These results suggest that the inhibitory effect of vandetanib on catecholamine synthesis in PC12 cells is mainly mediated by the inhibition of RET signaling.

### 2.3. Effects of Inhibition of RET-ERK and RET-AKT Pathways on Catecholamine Synthesis

RET is known to activate the Ras/Raf/MEK/ERK and PI3K/AKT signaling pathways [21]. To clarify the mechanism through which vandetanib suppresses catecholamine synthesis, we examined the effects of vandetanib on ERK and AKT activation by phosphorylation using Western blotting (Figure 5A,B and Appendix A). When PC12 cells were treated with 1000 nmol/L of vandetanib for 24 h, ERK and AKT phosphorylation decreased by 48.4% and 20.4%, respectively.

Next, we compared these two pathways to identify which pathway is key for the inhibition of catecholamine synthesis in PC12 cells. Cells were treated with 0.1 μmol/L of U0126, a Ras/Raf/MEK/ERK pathway inhibitor, or 0.1 μmol/L of LY294002, a PI3K/AKT pathway kinase inhibitor, for 24 h, and the catecholamine levels in cell lysates were assessed. Although the number of viable cells did not significantly change (Figure 6A and Appendix A), dopamine levels were reduced under treatment with U0126 and LY294002 by 36.0% and 24.8%, respectively (Figure 6B and Appendix A). Similarly, treatment with U0126 and LY294002 decreased noradrenaline levels by 29.1% and 20.0%, respectively (Figure 6C and Appendix A). These results prove that both Ras/Raf/MEK/ERK and PI3K/AKT pathways are important for catecholamine synthesis in PC12 cells.

## 3. Discussion

In this study, vandetanib, a multi-targeted TKI that has been clinically proven to be effective in RET-related cancers including MTC and NSCLC, suppressed the Ras/Raf/MEK/ERK and PI3K/AKT pathways by mainly inhibiting the tyrosine kinase activity of RET, leading to a decrease in catecholamine synthesis in PC12 cells.

Multi-targeted TKIs have been used to treat patients with advanced PPGLs. The therapeutic options for metastatic PPGLs remain limited. For unresectable PPGLs, radionuclide therapy with iodine-131 metaiodobenzylguanidine (MIBG) and chemotherapy with CVD (cyclophosphamide, vincristine, and dacarbazine) have been considered practiced standards of therapy. In addition to these two historically established options, antiangiogenic TKIs can be clinically used, with the primary aim of preventing activation of the most important angiogenesis regulator, the VEGF pathway [19]. Angiogenesis is known to be a hallmark of metastatic PPGL development, especially in cluster 1 PPGLs, since the pseudo-hypoxic mechanism caused by cluster 1 mutations leads to enhanced VEGF pathway signaling. To date, vandetanib and cabozantinib have been proven to be effective and are approved by the US Food and Drug Administration (FDA) and the European Medicines Agency (EMA) for the treatment of locally advanced and metastatic MTC [10]. In addition to these two drugs, two highly selective RET inhibitors, selpercatinib and prasetinib, are currently used in clinical practice for RET-related MTC [11,12]. Since most advanced PPGLs belong to cluster 1, treatment with TKIs is expected to inhibit the VEGF/VEGFR pathway and to suppress angiogenesis activated by the cluster 1 pseudo-hypoxic mechanism [10,14]. In contrast, our in vitro study showed that vandetanib could exert a direct inhibitory effect on catecholamine synthesis in tumor cells at lower drug concentrations than those affecting cell viability. Furthermore, knockdown of *RET* decreased intracellular catecholamine production in PC12 cells. PC12 cells do not express pathogenic variants of *RET* but do express such variant of another cluster 2 gene *MAX*. The MAX protein is a cofactor of the proto-oncogene MYC and mediates its function as a transcription factor via heterodimerization. Crosstalk between Ras/ERK or PI3K/AKT signaling and the MAX-MYC signaling pathway was previously reported [22]. Hence, alterations in MAX/MYC signaling are considered to be closely linked to the tumorigenesis caused by pathogenic variants of other cluster 2 genes including *RET* [23]. Accordingly, this gene variant of PC12 might have enhanced the effects of vandetanib observed in our study. Though most clinical trials that examined the antitumor effects of multi-targeted TKIs on metastatic PPGL did not include patients with MEN2-related PCCs [13,15], new clinical evidence regarding the effect of selpercatinib, a RET-specific TKI, on activated RET-related PPGLs has recently been reported [16,24]. Mweempwa et al. reported that a 66-year-old patient with metastatic PCC harboring a RET-SEPTIN9 fusion gene demonstrated a rapid response to selpercatinib: a 46% reduction in the sum of diameters of the target lesions within 12 weeks of treatment [24]. Deschler-Baier et al. reported on 6 PCC patients treated with selpercatinib in the LIBRETTO-001 study, a global phase 1/2 multicohort clinical trial involving 41 patients with solid tumors harboring an activating RET variant other than NSCLC and thyroid cancer [16]. Of the six PCC patients, one had a complete response, three had a partial response, and two had stable disease. Of note, serum metanephrine levels, which were elevated before the administration of selpercatinib, were markedly decreased even in the two patients with stable disease. The detailed mechanism through which selpercatinib regulates catecholamine secretion from PCC tumors remains unknown. The decrease in catecholamine secretion without tumor size reduction may be associated with the direct effect of the TKI on catecholamine synthesis, which was demonstrated in the present study using PC12 cells.

There are some limitations to this study. Though the suppressive effect of vandetanib on catecholamine synthesis was considered to be mainly mediated by weakened RET signaling, inhibitory effects on other potential off-targets of vandetanib could also be involved. Vandetanib is a broad-spectrum TKI; hence, selective RET inhibitors like selpercatinib may be more suitable to solve this problem. In addition, all of the experiments presented here were performed using a single cell line derived from rat PCC. Studies using the same protocol with other cell types, including human PPGL cell lines or primary tumor cells, which carry pathogenic *RET* variants or do not carry cluster 2-related gene mutations such as *RET* or *MAX*, should be performed to elucidate the influence of genetic background on the significance of RET signaling inhibition in relation to abnormal catecholamine production in PPGLs. PC12 cells do not express pathogenic variants of RET but do express such a variant of another cluster 2 gene *MAX*. Hence, *MAX* mutation in PC12 cells might have affected catecholamine synthesis or cellular signaling, the primary focal points of this study, compared with PPGLs with pathogenic *RET* variants or mutations of other cluster types.

## 4. Materials and Methods

### 4.1. Cell Culture

The rat pheochromocytoma cell line PC12 was purchased from the RIKEN Cell Bank (Tsukuba, Ibaraki, Japan). PC12 cells were cultured in a humidified atmosphere containing 5% CO_2_ at 37 °C in Dulbecco’s modified Eagle’s Medium (DMEM) supplemented with 10% fetal calf serum (FCS), 10% horse serum (HS), penicillin, and streptomycin purchased from Sigma-Aldrich (St. Louis, MO, USA). The culture medium was changed once per week, and the cells were passaged when grown to 80% confluence.

### 4.2. Transfection of Small Interfering RNA

PC12 cells were seeded in 12-well plates and transfected with small interfering RNA (siRNA) targeting *Ret*, *Flt4*, or *Egfr* using lipofectamine (Thermo Fisher Scientific, #13778030) according to the manufacturer’s protocol. These three siRNAs were purchased from Thermo Fisher Scientific (Waltham, MA, USA, s127686, s137502, and s127686, respectively). A nontargeting siRNA (Nippon GENE, Tokyo, Japan) was also transfected as a negative control. The knockdown efficiency for 48 h culture after transfection was confirmed by quantitative real-time polymerase chain reaction analysis as described below.

### 4.3. RNA Extraction and Quantitative Real-Time Polymerase Chain Reaction Analysis

PC12 cells were precultured in 12-well plates with DMEM supplemented with 10% FCS and 10% HS for 24 h. After removal of the medium, total cellular RNA was extracted using TRIzol (Invitrogen, Carlsbad, CA, USA). The extracted RNA (1 μg) was subjected to reverse transcription (RT) using ReverTra Ace^Ⓡ^ (Toyobo, Japan) according to the manufacturer’s protocol. All primer sequences are listed in Table 1. Aliquots of the polymerase chain reaction (PCR) products were electrophoresed on 1.5% agarose gels and visualized with fluorescent nucleic acid staining assays (GelRed, purchased from Biotium, Fremont, CA, USA). To assess the quantification of each target mRNA level, real-time PCR was performed using the QuantStudio real-time PCR system (Applied Biosystems, Waltham, MA, USA). The relative expression of each mRNA was determined using the ΔCt method, where ΔCt is the value obtained by subtracting the Ct value of ribosomal protein L19 (*Rpl19*) mRNA from that of the target mRNA [25]. The amount of target mRNA relative to *Rpl19* mRNA was expressed as 2-(ΔCt), and the results were expressed as the ratio of target mRNA to *Rpl19* mRNA.

### 4.4. Cell Viability Assay

Cell viability assays were performed using the CellTiter 96 AQueous One Solution Cell Proliferation Assay System (Promega, Madison, WI, USA) according to the manufacturer’s instructions. PC12 cells were precultured in each well of a 96-well plate, and the indicated concentrations of vandetanib (ChemScence, Monmouth Junction, NJ, USA, 08852), U0126 (MEK1/2 inhibitor; Promega, Madison, WI, USA, #V112A), and LY294002 (PI3K/AKT kinase inhibitor; Cell Signaling Technology, #9901) were added. After 24 h culture, PC12 cells were incubated with 20 μL of CellTiter 96 AQueous One Solution reagent^®^ per well. The absorbance at 490 nm was measured using a 96-well plate reader (Bio-Rad, Hercules, CA, USA, Model 680 XR).

### 4.5. Catecholamine Assay

PC12 cells were precultured in 12-well plates with DMEM containing 10% FCS and 10% HS for 24 h. The medium was then changed to DMEM containing 1% FCS and 1% HS, and the cells were treated with the indicated concentrations of vandetanib, U0126, and LY2940020 for 24 h. In the siRNA transfection protocol, 24 h after the transfection of siRNAs targeting the tyrosine kinase receptors, the medium was changed to DMEM containing 1% FCS and 1% HS. The cell lysate was collected after 24 h culture, and the levels of catecholamines were determined using high-performance liquid chromatography (HPLC; BML, Inc., Saitama, Japan) [25].

### 4.6. Western Blot Analysis

PC12 cells were pretreated with the indicated concentrations of vandetanib in DMEM. After 24 h culture, cells were solubilized, and the cell lysates were then subjected to sodium dodecyl sulfate–polyacrylamide gel electrophoresis (SDS-PAGE) followed by immunoblotting with a Phospho-p44/42 MAPK(Erk1/2) antibody (Cell Signaling Technology, Beverly, MA, USA, #9101, RRID: AB_331646), a p44/42 MAPK (Erk1/2) antibody (Cell Signaling Technology, #9102, RRID: AB_330744), a Phospho-Akt antibody (Cell Signaling Technology, #9271, RRID: AB_329825), an Akt antibody (Cell Signaling Technology, #9272, RRID: AB_329827), and an anti-ACTIN antibody (Sigma-Aldrich, A2066, RRID: AB_476693). The integrated signal density of each protein band was analyzed using the LAS-4000 mini-instrument (FUJIFILM, Tokyo, Japan).

### 4.7. Statistical Analysis

Data are presented as the means ± SE from at least three separate experiments, each performed in triplicate. Differences between groups were analyzed for statistical significance using ANOVA with a post hoc unpaired *t*-test, when appropriate, to determine differences. Two-tailed *p*-values less than 0.05 were regarded as statistically significant.

## 5. Conclusions

In conclusion, we have shown that the inhibition of RET signaling by vandetanib suppressed not only cell proliferation but also catecholamine synthesis via the Ras/Raf/MEK/ERK and PI3K/AKT pathways in PC12 cells. The silencing of mRNAs revealed that RET, rather than EGFR, is the potential target receptor for the inhibition of catecholamine synthesis in PC12 cells. These results suggest the possibility that RET-TKI may stabilize catecholamine excess in cluster 2 PPGLs, and future studies on catecholamine secretion control by TKIs are needed on human samples.

## Figures and Tables

**Figure 1 ijms-26-06927-f001:**
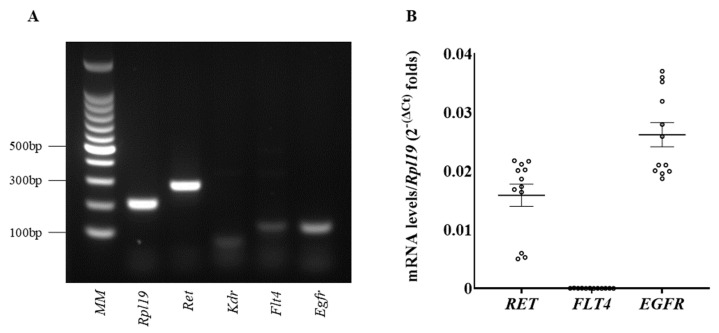
RNA expression of tyrosine kinase receptors in PC12 cells. (**A**) Total cellular RNAs were extracted from PC12 cells. The expression of Rpl19, a housekeeping gene; Ret; Flt4; and Egfr was confirmed by RT-PCR analysis, whereas that of Kdr was not detected. MM indicates a molecular weight marker. (**B**) Total cellular RNAs were extracted, and mRNA levels were analyzed by quantitative PCR. The expression levels of target mRNAs were standardized according to the Rpl19 level in the respective sample, and then the levels of mRNA of genes were expressed as fold changes. Data are shown as the mean ± SE from at least 3 separate experiments, each performed in triplicate.

**Figure 2 ijms-26-06927-f002:**
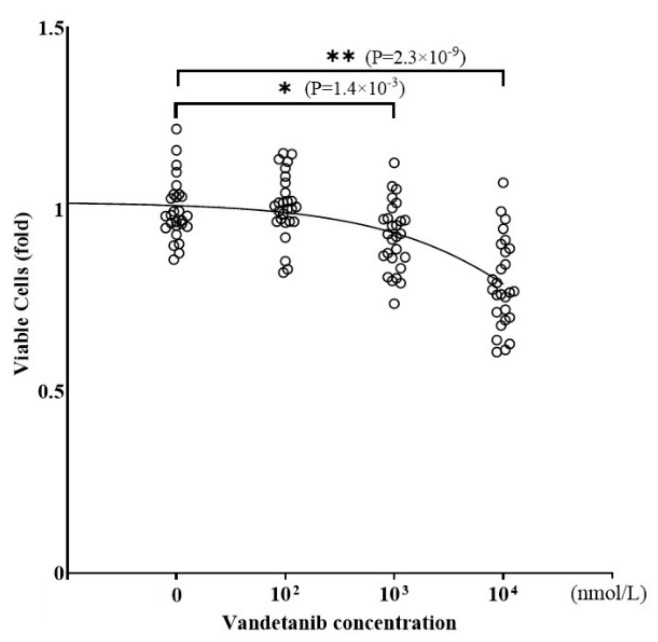
The effect of vandetanib on cell viability. Cells were seeded into 96-well culture plates at a density of 30,000 cells/well, then incubated for 24 h at 37 °C in a CO_2_ incubator with the indicated concentration of vandetanib. Viable cells were measured by CellTiter 96^®^. IC50: 110,568 nmol/L. Data were acquired from at least 3 separate experiments, each performed in triplicate. * *p* < 0.05 and ** *p* < 0.01 compared with the control as determined by an unpaired *t*-test.

**Figure 3 ijms-26-06927-f003:**
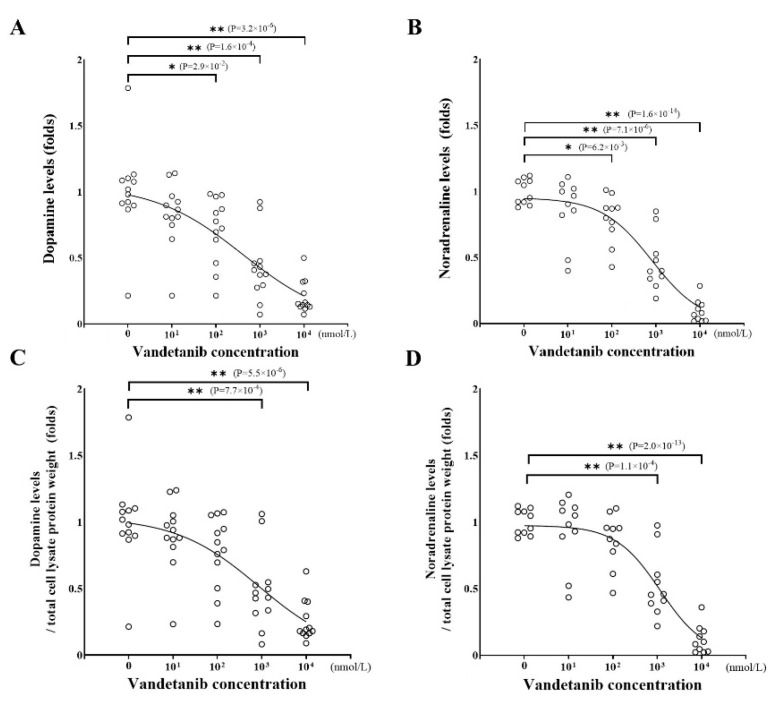
The effect of vandetanib on catecholamine synthesis. PC12 cells were cultured in DMEM containing 1% FCS and 1% HS, and then the cells were treated with the indicated concentrations of vandetanib. The cell lysate was collected after 24 h culture. (**A**) Dopamine levels in the cell lysate were measured by HPLC and expressed as fold changes standardized by each control level. IC50: 410 nmol/L. (**B**) Noradrenaline levels in the cell lysate were measured using the same method as for dopamine. IC50: 819.6 nmol/L. (**C**,**D**) The dopamine and noradrenaline levels shown in (**A**,**B**) were corrected for total protein weight in each cell lysate and expressed as fold changes standardized by each control level. (**C**) IC50: 855.8 nmol/L. (**D**) IC50: 1163 nmol/L. Data were acquired from at least 3 separate experiments, each performed in triplicate. * *p* < 0.05 and ** *p* < 0.01 compared with the control as determined by an unpaired *t*-test.

**Figure 4 ijms-26-06927-f004:**
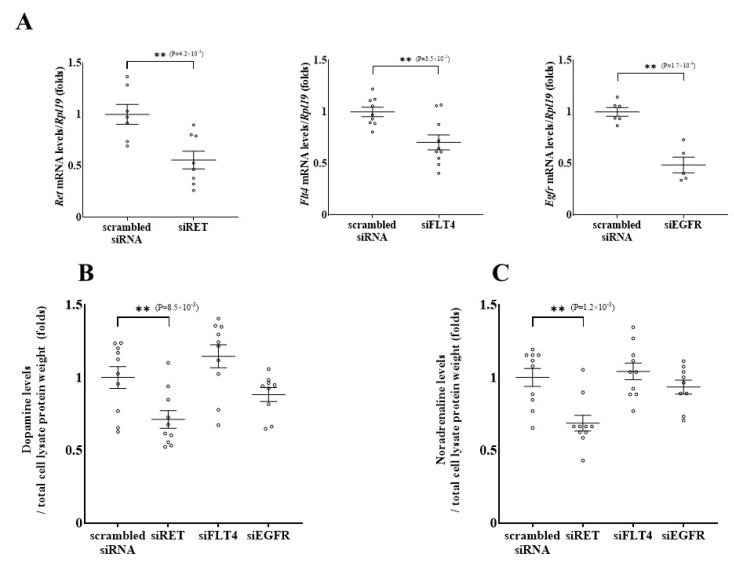
The effects of knockdown of the target receptor genes of vandetanib on catecholamine synthesis in PC12 cells. (**A**) The knockdown efficiency of siRNA-mediated gene silencing was confirmed by quantitative PCR. Total cellular RNAs were extracted 48 h after siRNA transfection, and mRNA levels of the target genes were analyzed by quantitative PCR. The expression levels were standardized according to the *Rpl19* level in the respective sample, and then the levels of mRNA of genes were expressed as fold changes. (**B**) The effect of knockdown of the tyrosine kinase receptors on dopamine synthesis in PC12 cells was evaluated. Dopamine levels in the cell lysate were measured by HPLC followed by correction for total protein weight in each cell lysate and expressed as fold changes standardized by each control level. (**C**) Noradrenaline levels in the cell lysate were measured using the same method as for dopamine. Data are shown as the mean ± SE from at least 3 separate experiments, each performed in triplicate. ** *p* < 0.01 compared with the control as determined by an unpaired *t*-test.

**Figure 5 ijms-26-06927-f005:**
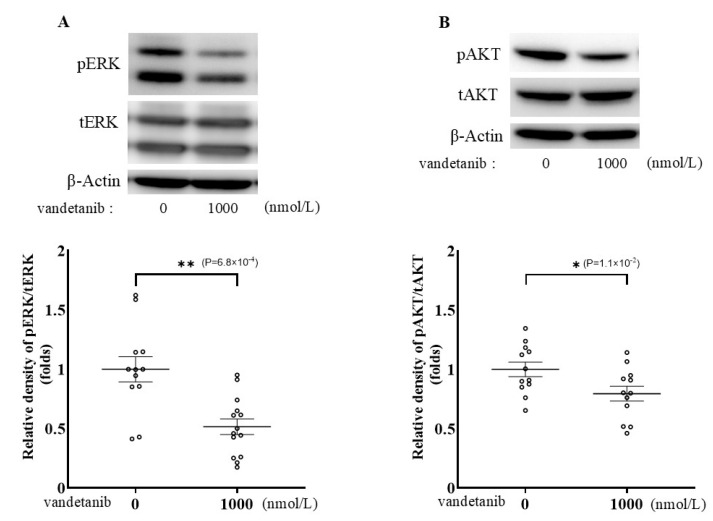
The effect of vandetanib on the phosphorylation of ERK and AKT. (**A**) After treatment with 1000 nmol/L of vandetanib for 24 h, the cell lysates were subjected to SDS-PAGE followed by immunoblotting with anti-phosphorylated ERK (pERK) and anti-total ERK (tERK) antibodies. (**B**) Immunoblotting with anti-phosphorylated AKT (pAKT) and anti-total AKT (tAKT) antibodies was performed using the same technique. Data are shown as the mean ± SE from at least 3 separate experiments, each performed in triplicate. * *p* < 0.05 and ** *p* < 0.01 compared with the control as determined by an unpaired *t*-test.

**Figure 6 ijms-26-06927-f006:**
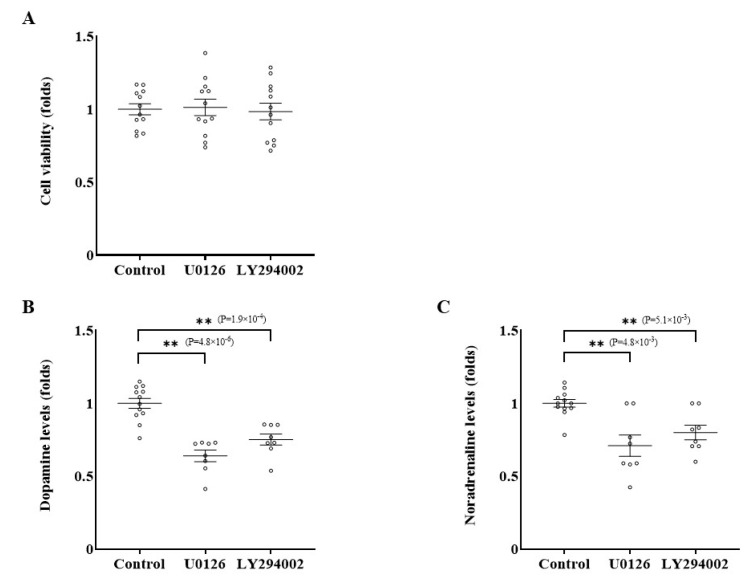
The effects of the inhibition of the ERK and AKT signaling pathways on catecholamine synthesis. (**A**) After treatment with U0126 or LY294002 for 24 h, the number of viable cells was measured using CellTiter 96^®^. (**B**) PC12 cells were treated with 0.1 μmol/L of U0126 or LY294002. The cell lysate was collected after 24 h culture. Dopamine levels in the cell lysate were measured by HPLC and expressed as fold changes standardized by each control level. (**C**) Noradrenaline levels in the cell lysate were measured using the same method as for dopamine. Data are shown as the mean ± SE from at least 3 separate experiments, each performed in triplicate. ** *p* < 0.01 compared with the control as determined by an unpaired *t*-test.

**Table 1 ijms-26-06927-t001:** Primer sequences used for PCR analysis.

Gene	Forward Primer (5′-3′)	Reverse Primer (5′-3′)
*Rpl19*	CTGAAGGTCAAAGGGAATGTG	GGACAGAGTCTTGATGATCTC
*Ret*	GTCCAGTCCAACAACAACTC	AGTTCTCCACGCAAACTTTC
*Kdr*	AAGTTGTTTGTCCAACATCTGG	CCGTCTTTTAGTACAATGCCTG
*Flt4*	GGACCTTGTCTGCTACAGTTTC	CAGTAAAATGTTCCGAGCAGC
*Egfr*	ACCTATCTGCACCATCGAC	TGGAGAATTCGAGAATCAACTC

## Data Availability

Some datasets generated and analyzed during the current study are not publicly available but are available from the corresponding author upon reasonable request.

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
