# Peer review of "Inhibitory Effects of Vandetanib on Catecholamine Synthesis in Rat Pheochromocytoma PC12 Cells"

_ijms, 2025, doi:10.3390/ijms26146927_

Round 1
Reviewer 1 Report
Comments and Suggestions for Authors
The article entitled “Inhibitory Effects of Vandetanib on Catecholamine Synthesis in Rat Pheochromocytoma PC12 Cells” is well structured, with clear experimental studies supporting the research’s idea, but the following points should be considered before accepting it for publication:
- Introduction: The introduction section shows a high level of similarity with existing literature. Rephrase the introduction to lower this level.
- Authors need to add more details to justify using vandetanib in their study among other TKIs.
- To improve the accessibility of the readers, briefly define the technical terms like “phosphatidylinositol 3-kinase (PI3K)/AKT pathway,” and “MEK inhibitor,” etc.
- Please cite the methodologies used in this article and indicate whether the reported method was followed exactly or modified in any way.
- The conclusion is too short. Add more details to summarize the main points and findings, emphasizing the research's significance.
Author Response
Comments and Suggestions for Authors
The article entitled “Inhibitory Effects of Vandetanib on Catecholamine Synthesis in Rat Pheochromocytoma PC12 Cells” is well structured, with clear experimental studies supporting the research’s idea, but the following points should be considered before accepting it for publication:
Introduction: The introduction section shows a high level of similarity with existing literature. Rephrase the introduction to lower this level.
Response: Thank you for pointing this out. We have rephrased and changed the Introduction based on your and other reviewers’ indications.
Authors need to add more details to justify using vandetanib in their study among other TKIs.
Response: We have added more details to highlight the use of vandetanib on line 69–81.
To improve the accessibility of the readers, briefly define the technical terms like “phosphatidylinositol 3-kinase (PI3K)/AKT pathway,” and “MEK inhibitor,” etc.
Response: Thank you for your valuable suggestion. We have unified the terminology of cell signaling and have provided the detailed information on line 191-192.
Please cite the methodologies used in this article and indicate whether the reported method was followed exactly or modified in any way.
Response: Thank you for your suggestion, and we had cited our previous article where the use of reported methods has been mentioned in the manuscript:
Morimoto, E.; Inagaki, K.; Komatsubara, M.; Terasaka, T.; Itoh, Y.; Fujisawa, S.; Sasaki, E.; Nishiyama, Y.; Hara, T.; Wada, J., Effects of Wnt-beta-Catenin Signaling and Sclerostin on the Phenotypes of Rat Pheochromocytoma PC12 Cells. J Endocr Soc 2022, 6, (10), bvac121.
The conclusion is too short. Add more details to summarize the main points and findings, emphasizing the research's significance.
Response: Thank you for pointing this out, and we have included more details of our works and have emphasized our research’s significance in the Conclusion section.

Reviewer 2 Report
Comments and Suggestions for Authors
1. The introduction lacks a clear hypothesis. State explicitly whether vandetanib’s effects on catecholamines are RET-specific or involve off-target mechanisms.
2. The link between RET inhibition and catecholamine synthesis is underexplored. Add prior evidence (or lack thereof) on TKIs modulating catecholamines.
3. Emphasize why catecholamine suppression matters (e.g., symptom relief in PPGLs).
4. Acknowledge early that PC12 cells lack RET mutations, which may limit translational relevance to human PCCs.
5. Vandetanib Concentrations-Justify the chosen doses (100–1000 nM). Are these clinically achievable? Compare to plasma levels in patients.
6. Provide data on off-target effects of siRNAs (e.g., scrambled siRNA controls).
7. Clarify why intracellular (not secreted) catecholamines were measured. Secretion assays would strengthen findings.
8. Pathway Inhibitors-U0126/LY294002 doses (0.1 µM) seem low—cite precedent or dose-response data to confirm selectivity.
9. Distinguish between anti-proliferative and cytotoxic effects. Use BrdU/CFSE for proliferation.
10. Figure 1-Include protein-level validation (Western blot) of RET/EGFR expression alongside mRNA.
11. Figure 2-Add IC50 values for vandetanib’s effects on viability.
12. Normalize catecholamines to cell number (not just protein) to rule out dilution effects.
13. RET Knockdown-Show rescue experiments (e.g., RET overexpression) to confirm specificity.
14. Pathway Data-Quantify pERK/pAKT inhibition at lower vandetanib doses (e.g., 100 nM) to match catecholamine effects.
15. Does vandetanib downregulate tyrosine hydroxylase (TH) or other catecholamine enzymes? Probe this.
16. Contrast with clinical reports where TKIs (e.g., sunitinib) increase catecholamines—discuss possible reasons.
17. Address how MAX mutations in PC12 cells may skew results vs. human RET-mutant PCCs.
18. Acknowledge that vandetanib’s EGFR/VEGFR inhibition could contribute (e.g., via hypoxia pathways).
19. Include full Western blots with molecular weight markers in supplements.
20. Add dose-response for U0126/LY294002 to confirm their selectivity at 0.1 µM.
21. Define all abbreviations at first use.
22. Fix minor errors (e.g., charachterized → characterized).
23. Improve transitions between sections (e.g., Results to Discussion).
24. Avoid implying clinical applicability without in vivo or patient-derived data.
25. Suggest validating findings in RET-mutant PCC models or patient samples.
26. Include recent studies on RET-specific TKIs (e.g., selpercatinib) in PPGLs.
Comments on the Quality of English Language
1. The introduction lacks a clear hypothesis. State explicitly whether vandetanib’s effects on catecholamines are RET-specific or involve off-target mechanisms.
2. The link between RET inhibition and catecholamine synthesis is underexplored. Add prior evidence (or lack thereof) on TKIs modulating catecholamines.
3. Emphasize why catecholamine suppression matters (e.g., symptom relief in PPGLs).
4. Acknowledge early that PC12 cells lack RET mutations, which may limit translational relevance to human PCCs.
5. Vandetanib Concentrations-Justify the chosen doses (100–1000 nM). Are these clinically achievable? Compare to plasma levels in patients.
6. Provide data on off-target effects of siRNAs (e.g., scrambled siRNA controls).
7. Clarify why intracellular (not secreted) catecholamines were measured. Secretion assays would strengthen findings.
8. Pathway Inhibitors-U0126/LY294002 doses (0.1 µM) seem low—cite precedent or dose-response data to confirm selectivity.
9. Distinguish between anti-proliferative and cytotoxic effects. Use BrdU/CFSE for proliferation.
10. Figure 1-Include protein-level validation (Western blot) of RET/EGFR expression alongside mRNA.
11. Figure 2-Add IC50 values for vandetanib’s effects on viability.
12. Normalize catecholamines to cell number (not just protein) to rule out dilution effects.
13. RET Knockdown-Show rescue experiments (e.g., RET overexpression) to confirm specificity.
14. Pathway Data-Quantify pERK/pAKT inhibition at lower vandetanib doses (e.g., 100 nM) to match catecholamine effects.
15. Does vandetanib downregulate tyrosine hydroxylase (TH) or other catecholamine enzymes? Probe this.
16. Contrast with clinical reports where TKIs (e.g., sunitinib) increase catecholamines—discuss possible reasons.
17. Address how MAX mutations in PC12 cells may skew results vs. human RET-mutant PCCs.
18. Acknowledge that vandetanib’s EGFR/VEGFR inhibition could contribute (e.g., via hypoxia pathways).
19. Include full Western blots with molecular weight markers in supplements.
20. Add dose-response for U0126/LY294002 to confirm their selectivity at 0.1 µM.
21. Define all abbreviations at first use.
22. Fix minor errors (e.g., charachterized → characterized).
23. Improve transitions between sections (e.g., Results to Discussion).
24. Avoid implying clinical applicability without in vivo or patient-derived data.
25. Suggest validating findings in RET-mutant PCC models or patient samples.
26. Include recent studies on RET-specific TKIs (e.g., selpercatinib) in PPGLs.
Author Response
Comments and Suggestions for Authors
- The introduction lacks a clear hypothesis. State explicitly whether vandetanib’s effects on catecholamines are RET-specific or involve off-target mechanisms.
Response: Thank you for pointing this out. We have added a clearer hypothesis on catecholamine synthesis in this study on vandetanib. Silencing of Ret leads to the decrease in catecholamine synthesis; however, the off-target mechanism of vandetanib was not fully inhibited because vandetanib is a multi-target TKI. - The link between RET inhibition and catecholamine synthesis is underexplored. Add prior evidence (or lack thereof) on TKIs modulating catecholamines.
Response: Thank you for pointing this out. They were clinical data, but recently the RET-specific inhibitor selpercatinib was also reported to decrease catecholamine levels without tumor size reduction. We have described this on line 68–74 in the revised manuscript. We think that more evidence for the direct inhibition of catecholamine production in vitro should be examined as described by us in this article. - Emphasize why catecholamine suppression matters (e.g., symptom relief in PPGLs).
Response: We have described the importance of catecholamine reduction in clinical PPGL patients on line 76–77. - Acknowledge early that PC12 cells lack RET mutations, which may limit translational relevance to human PCCs.
Response: We have included the information on the lack of RET mutations in PC12 cells on lines 79–81 and 271–274. - Vandetanib Concentrations-Justify the chosen doses (100–1000 nM). Are these clinically achievable? Compare to plasma levels in patients.
Response: The Cmax (mean ± standard deviation) and its conversion to nmol/L for 100, 200, 300, and 400 mg of vandetanib (Caprelsa®) administered orally once daily for 28 days to Japanese patients with solid tumors are shown below. The vandetanib concentrations used in this experiment did not deviate significantly from the clinical dosage.
Vandetanib (dose 100 mg/day, Cmax 1200±583 ng/mL, 1297.9–3750.9 nmol/L: dose 200 mg/day, Cmax 922±258 ng/mL, 1396.8–2482.3 nmol/L: dose 300 mg/day, Cmax 1580±302, 2688.5–3959.1 nmol/L, dose 400 mg/day, Cmax 2050 ng/ml, 4312.6 nmol/L).
- Provide data on off-target effects of siRNAs (e.g., scrambled siRNA controls).
Response: We have followed your advice and the figures have been changed for the negative control, i.e., the scrambled siRNA control (the negative control is the scrambled siRNA transfected control), in the revised manuscript. - Clarify why intracellular (not secreted) catecholamines were measured. Secretion assays would strengthen findings.
Response: We had measured secreted catecholamine in the cultured media as well; however, the results were not consistent enough and the noradrenaline levels were also too low to be analyzed. - Pathway Inhibitors-U0126/LY294002 doses (0.1 µM) seem low—cite precedent or dose-response data to confirm selectivity.
Response: We have included the dose–response curve of the cell viability assay and dopamine and noradrenaline assay in the Supplementary Figure (Fig. S8). The cell viability was slightly decreased at 1 µM of both U0126 and LY294002, and catecholamine reduction was sufficiently detected at 0.1 µM of both U0126 and LY294002. This is the reason why we had chosen a 0.1 µM concentration for both U0126 and LY294002. - Distinguish between anti-proliferative and cytotoxic effects. Use BrdU/CFSE for proliferation.
Response: We did not use BrdU/CFSE for proliferation, and we could only distinguish cell viability and not anti-proliferative or cytotoxic effects by the CellTitre assay in this model. - Figure 1-Include protein-level validation (Western blot) of RET/EGFR expression alongside mRNA.
Response: We did not validate the RET/EGFR protein in PC12 cells, but the following research article has assessed the RET/EGFR expressions of PC12 cells by the WB analysis. RET: Journal of Biological Chemistry 2002; 277, 43623-63630. DOI: 10.1074/jbc.M203926200, EGFR: Horm Metab Res 2009; 41: 710 – 714. DOI 10.1055/s-0029-1224136. - Figure 2-Add IC50 values for vandetanib’s effects on viability.
Response: Thank you for pointing this out. We have included the IC50 values as per your recommendation in the revised manuscript. - Normalize catecholamines to cell number (not just protein) to rule out dilution effects.
Response: When we collect the protein samples of PC12 cells, we do not count the cell numbers, and we usually use the protein levels to normalize hormone production because we directly collect protein samples by using RIPA buffer for dissolution. We do not think using this step causes any dilution effects. - RET Knockdown-Show rescue experiments (e.g., RET overexpression) to confirm specificity.
Response: Unfortunately, we did not use the RET overexpression model. - Pathway Data-Quantify pERK/pAKT inhibition at lower vandetanib doses (e.g., 100 nM) to match catecholamine effects.
Response: We had also conducted the experimented at a lower vandetanib dose of 100 nM, and the inhibition of phosphorylation was detected, but it was not clear enough to be included in the publication. - Does vandetanib downregulate tyrosine hydroxylase (TH) or other catecholamine enzymes? Probe this.
Response: We had assessed TH and other catecholamine enzyme expression levels, but their expression levels were unchanged. For the decrease in catecholamine synthesis, the phosphorylation of catecholamine enzymes could be the cause for the vandetanib effect, but we have not proved it yet. - Contrast with clinical reports where TKIs (e.g., sunitinib) increase catecholamines—discuss possible reasons.
Response: As you pointed out, a few reports have discussed the increase in catecholamine levels by TKIs, but PPGL patient studies have revealed that catecholamine levels decrease without PPGL volume reduction in sunitinib or selpercatinib treatment. Please refer to Refs. 16 and 17 in the revised manuscript. - Address how MAX mutations in PC12 cells may skew results vs. human RET-mutant PCCs.
Response: We had addressed this problem on lines 239–245 and 267–274 in the revised manuscript. - Acknowledge that vandetanib’s EGFR/VEGFR inhibition could contribute (e.g., via hypoxia pathways).
Response: Thank you for pointing this out. Silencing of EGFR or VEGFR (Flt4) did not affect catecholamine synthesis in PC12 cells in this study, and we concluded that catecholamine reduction by vandetanib should be via RET in PC12 cells. We have discussed this on line 152–157 in the revised manuscript. - Include full Western blots with molecular weight markers in supplements.
Response: We have included the molecular weight markers in Supplement Figures. - Add dose-response for U0126/LY294002 to confirm their selectivity at 0.1 µM.
Response: We have included the dose–response curve of cell the viability assay and dopamine and noradrenaline assay in the Supplementary Figure (Fig. S8). The cell viability was slightly decreased at 1 µM of both U0126 and LY294002, and catecholamine reduction was sufficiently detected at 0.1 µM of both U0126 and LY294002. This is the reason why we had chosen a concentration of 0.1 µM for both U0126 and LY294002. - Define all abbreviations at first use.
Response: Thank you for pointing this out, and we have defined all abbreviations at their first use in the revised manuscript. - Fix minor errors (e.g., charachterized → characterized).
Response: We have fixed this in the revised manuscript. - Improve transitions between sections (e.g., Results to Discussion).
Response: Thank you for pointing this out. We have improved the transition in the best way possible in the revised manuscript. - Avoid implying clinical applicability without in vivo or patient-derived data.
Response: Thank you for pointing this out. We have avoided implying the direct clinical applicability to PPGL patients in the revised manuscript. - Suggest validating findings in RET-mutant PCC models or patient samples.
Response: We have included this suggestion on line 261–265 of the revised manuscript. - Include recent studies on RET-specific TKIs (e.g., selpercatinib) in PPGLs.
Response: We have included the recent studies on RET-specific TKIs in the revised manuscript.

Reviewer 3 Report
Comments and Suggestions for Authors
This manuscript addresses the potential direct effects of vandetanib, a multi-target tyrosine kinase inhibitor (TKI), on catecholamine synthesis in PC12 cells, a model of pheochromocytoma (PCC). The authors systematically explore RET signaling’s role in catecholamine production and examine downstream pathways (ERK, AKT). This is an interesting and relevant study, especially in light of emerging selective RET inhibitors and limited treatment options for PPGLs. However, the manuscript requires significant revisions to improve clarity, precision of language, and presentation. Numerous issues with English expression, grammar, and manuscript organization detract from readability. Furthermore, some aspects of experimental design and data interpretation warrant clarification. I do not recommend rejection, but strongly advise major revisions (listed below) before the manuscript is suitable for publication.
- Limited Model System (Use of Only PC12 Cells): A major limitation of the study is that all experiments were performed exclusively in the rat pheochromocytoma PC12 cell line. While PC12 cells are an established model for studying catecholamine synthesis, they may not fully recapitulate the genetic and phenotypic diversity of human PPGLs, particularly those harboring specific RET mutations or other cluster 2 alterations.
I recommend that the authors:
(A) Justify their choice to use only PC12 cells.
(B) Discuss the limitations of relying on a single rat cell line and how differences between PC12 cells and human PPGLs might affect the generalizability of their findings.
(C) Indicate whether other models (e.g., human PPGL cell lines or primary tumor cells) were considered or are available for future validation. - Several figures in the manuscript, including those displaying catecholamine levels, cell viability, and signaling pathway analyses, are currently shown as bar graphs with mean ± SE for each treatment condition. While this format is acceptable, it limits the reader’s ability to evaluate data distribution and experimental variability.
I recommend the following improvements:
(A) Where possible, display individual data points (e.g., scatter or dot plots overlaid on bars) to illustrate variability across replicates.
(B) For dose-response experiments (e.g., Figure 2 cell viability data), replot using nonlinear regression curves rather than discrete bars to better visualize drug effects and calculate parameters such as IC50 values.
(C) Ensure consistent axis scales and labels across figures to facilitate comparisons.
(D) Use appropriate statistical annotations (e.g., exact P-values, statistical test used) on all graphs.
(E) Ensure that curve fitting details (e.g., model used, software) are described in the Methods section. - The current title describes the study’s focus but does not adequately convey the mechanistic insights into RET signaling and its downstream pathways (ERK, AKT) that are central to the manuscript. A more precise and informative title would help readers immediately grasp the study’s significance and mechanistic contribution.
- The abstract in its current form reads somewhat scattered and unfocused, and needs improvement. It blends background information with results in long, complex sentences and lacks clear structure.
Specific issues I noted:
(A) Organization: The abstract should follow a clear structure: brief background, objective/hypothesis, methods, key results (with specific quantitative findings), and conclusions.
(B) Language and Grammar: There are numerous instances of awkward phrasing and grammatical errors (e.g., overly long sentences, and misused words).
(C) Abbreviations: Several abbreviations appear without definition (e.g., PPGL) or are used inconsistently.
(D) Coherence: Transitions between background, methods, and results are abrupt, leading to a sense of disconnected statements rather than a unified narrative. - Although the introduction contains relevant information, it is excessively long, at times unfocused, and includes unnecessary detail that detracts from the manuscript’s clarity. Repeated background on RET signaling and lengthy discussions of various cancers and treatment regimens obscure the main purpose of the study. Additionally, certain details (e.g., specific patient case reports of selpercatinib responses) are not appropriate for the introduction and belong in the discussion, where they can be critically evaluated in context.
I recommend that the authors:
(A) Condense and reorganize the introduction to remove redundant information and maintain focus on the specific scientific question addressed by this study.
(B) Retain only the essential background needed to understand the rationale and significance of investigating vandetanib’s effects on catecholamine synthesis.
(C) Eliminate or relocate overly detailed clinical examples and treatment histories to the discussion.
(D) End the introduction with a precise, concise statement of the study’s hypothesis or objectives. - The results need sufficient depth and interpretation: While the results section is organized under clear subheadings and presents the experimental findings, the discussion of the results is too brief, vague, and narrowly focused on reporting numerical outcomes without sufficient interpretation or context. Many observations are stated as facts without a deeper explanation of their biological or mechanistic implications.
I recommend that the authors:
(A) Expand their commentary within the results section to briefly interpret the significance of each major finding rather than merely reporting numerical values.
(B) Discuss how their findings compare with or differ from previous studies, especially regarding RET signaling’s role in catecholamine synthesis.
(C) Highlight novel or unexpected results (e.g., the differential concentration effect on catecholamine synthesis vs. cell viability) and speculate on possible mechanisms.
(D) Avoid purely descriptive language and aim to connect data to broader scientific questions or potential clinical relevance. - The study includes an assessment of EGFR mRNA expression in PC12 cells and knockdown experiments targeting EGFR. However, the manuscript does not clearly explain the biological rationale for investigating EGFR in the context of catecholamine synthesis or PPGL pathophysiology. While EGFR can be a target of vandetanib, the link to catecholamine production in this model remains unexplored.
I recommend that the authors:
(A) Provide a clear explanation of why EGFR was included in their experimental design.
(B) Interpret why EGFR knockdown did not influence catecholamine synthesis, and whether this result was expected or surprising. - The discussion provides a reasonable summary of the findings and places them in context. However, there is significant room for improvement in depth, clarity, and critical analysis. In its current form, the discussion repeats background information from the introduction rather than focusing on interpreting the novel findings of this study.
I recommend that the authors:
(A) Reduce repetition of background content and instead devote more space to interpreting the meaning and implications of their results.
(B) Provide a more critical analysis of the study’s limitations, particularly the use of only one cell line and the lack of human cell models.
(C) Elaborate on the potential clinical significance of the finding that catecholamine synthesis is inhibited at lower drug concentrations than those affecting cell viability, which could be therapeutically relevant.
(D) Discuss possible mechanistic explanations for their observations, including potential off-target effects of vandetanib and how RET inhibition may impact catecholamine synthesis pathways.
(E) Suggest clear future directions for research to validate these findings in human cell-based systems or in vivo models. - The conclusion section, while summarizing the study’s primary findings, is brief and lacks sufficient emphasis on the broader significance and potential impact of the work. As written, it reads more like a restatement of results rather than a synthesis that conveys the importance of the findings.
I recommend that the authors:
(A) Expand the conclusion to highlight the novel contributions of this study to the field.
(B) Emphasize how their findings might inform future research or potential therapeutic approaches for PPGLs.
(C) Avoid simply repeating results and instead focus on the implications and next steps arising from the work.
Overall, I find the manuscript abrupt and lacking in cohesive narrative flow. The transitions between sections (Introduction, Results, Discussion) feel disjointed, and individual paragraphs are often read as standalone statements rather than parts of a unified scientific story. As a result, the manuscript does not effectively guide the reader through the logical progression of the research question, experimental findings, and broader significance. Beyond issues of narrative flow, I find that the experimental design lacks sufficient explanation and justification. The rationale for certain experiments, choice of concentrations, and selection of targets (e.g., EGFR) is not always clear, and the logic connecting different experimental steps sometimes appears fragmented. As a result, the overall design feels incomplete and disconnected from the central research question.
Author Response
Comments and Suggestions for Authors
This manuscript addresses the potential direct effects of vandetanib, a multi-target tyrosine kinase inhibitor (TKI), on catecholamine synthesis in PC12 cells, a model of pheochromocytoma (PCC). The authors systematically explore RET signaling’s role in catecholamine production and examine downstream pathways (ERK, AKT). This is an interesting and relevant study, especially in light of emerging selective RET inhibitors and limited treatment options for PPGLs. However, the manuscript requires significant revisions to improve clarity, precision of language, and presentation. Numerous issues with English expression, grammar, and manuscript organization detract from readability. Furthermore, some aspects of experimental design and data interpretation warrant clarification. I do not recommend rejection, but strongly advise major revisions (listed below) before the manuscript is suitable for publication.
Limited Model System (Use of Only PC12 Cells): A major limitation of the study is that all experiments were performed exclusively in the rat pheochromocytoma PC12 cell line. While PC12 cells are an established model for studying catecholamine synthesis, they may not fully recapitulate the genetic and phenotypic diversity of human PPGLs, particularly those harboring specific RET mutations or other cluster 2 alterations.
I recommend that the authors:
(A) Justify their choice to use only PC12 cells.
(B) Discuss the limitations of relying on a single rat cell line and how differences between PC12 cells and human PPGLs might affect the generalizability of their findings.
(C) Indicate whether other models (e.g., human PPGL cell lines or primary tumor cells) were considered or are available for future validation.
Response: Thank you for your important pointing out. We put the limitation on only PC12 use in introduction (line 78-81) and in discussion (line 267-274) in the revision. We agree that further studies with other model (especially by human samples) are needed for the validation or confirmation.
Several figures in the manuscript, including those displaying catecholamine levels, cell viability, and signaling pathway analyses, are currently shown as bar graphs with mean ± SE for each treatment condition. While this format is acceptable, it limits the reader’s ability to evaluate data distribution and experimental variability.
I recommend the following improvements:
(A) Where possible, display individual data points (e.g., scatter or dot plots overlaid on bars) to illustrate variability across replicates.
(B) For dose-response experiments (e.g., Figure 2 cell viability data), replot using nonlinear regression curves rather than discrete bars to better visualize drug effects and calculate parameters such as IC50 values.
(C) Ensure consistent axis scales and labels across figures to facilitate comparisons.
(D) Use appropriate statistical annotations (e.g., exact P-values, statistical test used) on all graphs.
(E) Ensure that curve fitting details (e.g., model used, software) are described in the Methods section.
Response: Thank you for pointing this out. We have added the nonlinear regression curves and IC50 values for the dose–response experiments in the Fig. 2 and Fig. 3 charts in the revised manuscript. Furthermore, we have added the consistent axis scales in the same figures for comparisons. Also, we have included the exact P-values on all graphs. The statistical methods are provided in the Methods section.
The current title describes the study’s focus but does not adequately convey the mechanistic insights into RET signaling and its downstream pathways (ERK, AKT) that are central to the manuscript. A more precise and informative title would help readers immediately grasp the study’s significance and mechanistic contribution.
The abstract in its current form reads somewhat scattered and unfocused, and needs improvement. It blends background information with results in long, complex sentences and lacks clear structure.
Specific issues I noted:
(A) Organization: The abstract should follow a clear structure: brief background, objective/hypothesis, methods, key results (with specific quantitative findings), and conclusions.
(B) Language and Grammar: There are numerous instances of awkward phrasing and grammatical errors (e.g., overly long sentences, and misused words).
(C) Abbreviations: Several abbreviations appear without definition (e.g., PPGL) or are used inconsistently.
(D) Coherence: Transitions between background, methods, and results are abrupt, leading to a sense of disconnected statements rather than a unified narrative.
Response: Thank you for pointing this out. We have changed the Abstract based on your suggestions.
Although the introduction contains relevant information, it is excessively long, at times unfocused, and includes unnecessary detail that detracts from the manuscript’s clarity. Repeated background on RET signaling and lengthy discussions of various cancers and treatment regimens obscure the main purpose of the study. Additionally, certain details (e.g., specific patient case reports of selpercatinib responses) are not appropriate for the introduction and belong in the discussion, where they can be critically evaluated in context.
I recommend that the authors:
(A) Condense and reorganize the introduction to remove redundant information and maintain focus on the specific scientific question addressed by this study.
(B) Retain only the essential background needed to understand the rationale and significance of investigating vandetanib’s effects on catecholamine synthesis.
(C) Eliminate or relocate overly detailed clinical examples and treatment histories to the discussion.
(D) End the introduction with a precise, concise statement of the study’s hypothesis or objectives.
Response: Thank you for your pointing out to the introduction. We condensed for the essential background and relocated some redundant information in discussion in the revision.
The results need sufficient depth and interpretation: While the results section is organized under clear subheadings and presents the experimental findings, the discussion of the results is too brief, vague, and narrowly focused on reporting numerical outcomes without sufficient interpretation or context. Many observations are stated as facts without a deeper explanation of their biological or mechanistic implications.
I recommend that the authors:
(A) Expand their commentary within the results section to briefly interpret the significance of each major finding rather than merely reporting numerical values.
(B) Discuss how their findings compare with or differ from previous studies, especially regarding RET signaling’s role in catecholamine synthesis.
(C) Highlight novel or unexpected results (e.g., the differential concentration effect on catecholamine synthesis vs. cell viability) and speculate on possible mechanisms.
(D) Avoid purely descriptive language and aim to connect data to broader scientific questions or potential clinical relevance.
Response: Thank you for valuable suggestion regarding the Results section. We have added the brief interpretation of each finding in the Results section and have highlighted the differential concentration effect on catecholamine synthesis vs. cell viability in the revised manuscript.
The study includes an assessment of EGFR mRNA expression in PC12 cells and knockdown experiments targeting EGFR. However, the manuscript does not clearly explain the biological rationale for investigating EGFR in the context of catecholamine synthesis or PPGL pathophysiology. While EGFR can be a target of vandetanib, the link to catecholamine production in this model remains unexplored.
I recommend that the authors:
(A) Provide a clear explanation of why EGFR was included in their experimental design.
(B) Interpret why EGFR knockdown did not influence catecholamine synthesis, and whether this result was expected or surprising.
Response: Thank you for pointing this out. We have added a clear explanation to include EGFR in the experimental design and the interpretation of no effect of EGFR on catecholamine synthesis on line 152–157.
The discussion provides a reasonable summary of the findings and places them in context. However, there is significant room for improvement in depth, clarity, and critical analysis. In its current form, the discussion repeats background information from the introduction rather than focusing on interpreting the novel findings of this study.
I recommend that the authors:
(A) Reduce repetition of background content and instead devote more space to interpreting the meaning and implications of their results.
(B) Provide a more critical analysis of the study’s limitations, particularly the use of only one cell line and the lack of human cell models.
(C) Elaborate on the potential clinical significance of the finding that catecholamine synthesis is inhibited at lower drug concentrations than those affecting cell viability, which could be therapeutically relevant.
(D) Discuss possible mechanistic explanations for their observations, including potential off-target effects of vandetanib and how RET inhibition may impact catecholamine synthesis pathways.
(E) Suggest clear future directions for research to validate these findings in human cell-based systems or in vivo models.
Response: Following your instruction, we have revised the Discussion section.
The conclusion section, while summarizing the study’s primary findings, is brief and lacks sufficient emphasis on the broader significance and potential impact of the work. As written, it reads more like a restatement of results rather than a synthesis that conveys the importance of the findings.
I recommend that the authors:
(A) Expand the conclusion to highlight the novel contributions of this study to the field.
(B) Emphasize how their findings might inform future research or potential therapeutic approaches for PPGLs.
(C) Avoid simply repeating results and instead focus on the implications and next steps arising from the work.
Response: Following your instruction, we have revised the Conclusion section.
Overall, I find the manuscript abrupt and lacking in cohesive narrative flow. The transitions between sections (Introduction, Results, Discussion) feel disjointed, and individual paragraphs are often read as standalone statements rather than parts of a unified scientific story. As a result, the manuscript does not effectively guide the reader through the logical progression of the research question, experimental findings, and broader significance. Beyond issues of narrative flow, I find that the experimental design lacks sufficient explanation and justification. The rationale for certain experiments, choice of concentrations, and selection of targets (e.g., EGFR) is not always clear, and the logic connecting different experimental steps sometimes appears fragmented. As a result, the overall design feels incomplete and disconnected from the central research question.

Round 2
Reviewer 2 Report
Comments and Suggestions for Authors
Authors revised all my queries. I accept this manuscript in current form.
Author Response
Thank you for your valuable suggestions of review.
Reviewer 3 Report
Comments and Suggestions for Authors
I have carefully reviewed the revised manuscript. The authors have adequately addressed the corrections and clarifications raised during the initial review. The experimental data are now well-organized, and the mechanistic insights into RET signaling are clearly presented. However, I have the following minor remaining points for the authors to address:
- In the abstract, the rationale for choosing Vandetanib requires a clearer explanation. Currently, the text jumps from discussing other TKIs (sunitinib, selpercatinib) to testing Vandetanib without explicitly connecting why Vandetanib was selected for this study. Please briefly clarify in the abstract that Vandetanib is a clinically used RET inhibitor for MTC and potentially relevant to RET-driven PPGLs.
- While the current cell viability data are acceptable for this manuscript, the tested concentration range is limited. For future work, I encourage the authors to evaluate a broader spectrum of concentrations, ideally using an 8-point dose–response curve (2-fold or 3-fold), which is typically accepted for defining IC50 values robustly and assessing potential off-target effects.
- For complete transparency and reproducibility, I recommend that raw numerical data files for all the assays be submitted to the journal as supplementary material or deposited in a suitable data repository.
These issues are minor and do not affect my overall conclusion. The manuscript is now scientifically sound, well-written, and contributes valuable insights into the direct effects of RET inhibition on catecholamine synthesis in PCC/PPGL models. I recommend acceptance after minor revision.
Author Response
I have carefully reviewed the revised manuscript. The authors have adequately addressed the corrections and clarifications raised during the initial review. The experimental data are now well-organized, and the mechanistic insights into RET signaling are clearly presented. However, I have the following minor remaining points for the authors to address:
In the abstract, the rationale for choosing Vandetanib requires a clearer explanation. Currently, the text jumps from discussing other TKIs (sunitinib, selpercatinib) to testing Vandetanib without explicitly connecting why Vandetanib was selected for this study. Please briefly clarify in the abstract that Vandetanib is a clinically used RET inhibitor for MTC and potentially relevant to RET-driven PPGLs.
Response: Thank you for pointing this out. We have rephrased and changed the abstract based on your indications.
While the current cell viability data are acceptable for this manuscript, the tested concentration range is limited. For future work, I encourage the authors to evaluate a broader spectrum of concentrations, ideally using an 8-point dose–response curve (2-fold or 3-fold), which is typically accepted for defining IC50 values robustly and assessing potential off-target effects.
Response: Thank you for your valuable suggestions. Unfortunately, we have not conducted broader spectrum tests to perform the analysis of IC50 as you pointed out in. We agree that further studies are needed for the confirmation.
For complete transparency and reproducibility, I recommend that raw numerical data files for all the assays be submitted to the journal as supplementary material or deposited in a suitable data repository.
Response: Thank you for pointing this out. We have included the raw numerical PDF data file (Raw data) for all the assays.
These issues are minor and do not affect my overall conclusion. The manuscript is now scientifically sound, well-written, and contributes valuable insights into the direct effects of RET inhibition on catecholamine synthesis in PCC/PPGL models. I recommend acceptance after minor revision.